# Rényi Entropies of Multidimensional Oscillator and Hydrogenic Systems with Applications to Highly Excited Rydberg States

**DOI:** 10.3390/e24111590

**Published:** 2022-11-02

**Authors:** Jesús S. Dehesa

**Affiliations:** 1Instituto Carlos I de Física Teórica y Computacional, Universidad de Granada, 18071 Granada, Spain; dehesa@ugr.es; 2Departamento de Física Atómica, Molecular y Nuclear, Universidad de Granada, 18071 Granada, Spain

**Keywords:** Rényi entropy inequalities, Rényi entropies of multidimensional oscillator systems, Rényi entropies of multidimensional hydrogenic systems, Rényi entropies of highly excited Rydberg states, hypergeometric orthogonal polynomials, asymptotics of Hermite, Laguerre and Gegenbauer polynomials, 03.65.-w: 02.30.Gp, 02.30.Mv, 03.65.Fd, 03.67.-a, 05.20.-y, 05.30.-d, 032.80.Ee

## Abstract

The various facets of the internal disorder of quantum systems can be described by means of the Rényi entropies of their single-particle probability density according to modern density functional theory and quantum information techniques. In this work, we first show the lower and upper bounds for the Rényi entropies of general and central-potential quantum systems, as well as the associated entropic uncertainty relations. Then, the Rényi entropies of multidimensional oscillator and hydrogenic-like systems are reviewed and explicitly determined for all bound stationary position and momentum states from first principles (i.e., in terms of the potential strength, the space dimensionality and the states’s hyperquantum numbers). This is possible because the associated wavefunctions can be expressed by means of hypergeometric orthogonal polynomials. Emphasis is placed on the most extreme, non-trivial cases corresponding to the highly excited Rydberg states, where the Rényi entropies can be amazingly obtained in a simple, compact, and transparent form. Powerful asymptotic approaches of approximation theory have been used when the polynomial’s degree or the weight-function parameter(s) of the Hermite, Laguerre, and Gegenbauer polynomials have large values. At present, these special states are being shown of increasing potential interest in quantum information and the associated quantum technologies, such as e.g., quantum key distribution, quantum computation, and quantum metrology.

## 1. Introduction

The Rényi entropies [1,2] for the probability densities ρ(r) and γ(p) are the most appropriate measures of quantum uncertainty, a property of fundamental relevance for the precise characterization of the position and momentum single-particle densities of quantum systems and the basic variables of the modern density-functional theory [3,4] in the two complementary spaces. For *D*-dimensional systems, they are
(1)$$R_{q}\left[\rho \right]=\frac{1}{1-q}\ln\int_{R_{D}}{\left[\rho \left(r\right)\right]}^{q}\;dr,\;0<q<\infty ,\;\;q\neq 1,$$
and
(2)$$R_{q^{*}}\left[\gamma \right]=\frac{1}{1-q^{*}}\ln\int_{R_{D}}{\left[\gamma \left(p\right)\right]}^{q^{*}}\;dp,\;0<q^{*}<\infty ,\;\;q^{*}\neq 1,$$
respectively. They supply a family of entropic measures of quantum states, depending on a real parameter *q*. This order parameter controls the concentration of the probability density over different regions of the hyperspace. Higher values of *q* indicate that the function ${\left[\rho \left(r\right)\right]}^{q}$ is more concentrated around the local maxima of the distribution, while the lower values have the effect of smoothing that functions over its whole domain of definition. The Rényi entropies quantify a great deal of spreading facets of ρ(r) over $R_{D}$, and describe numerous information-theoretical quantities as special cases, such as, e.g., the disequilibrium $D\left[\rho \right]=\exp(-R_{2}\left[\rho \right])$ and the Shannon entropy $S\left[\rho \right]=\lim_{q\to 1}R_{q}\left[\rho \right]$[5]. The Tsallis entropies $T_{q}\left[\rho \right]=\frac{1}{1-q}\left[q^{\left(1-q\right)R_{q}\left[\rho \right]}-1\right]$ [6,7] and various generalized complexity measures [8,9,10,11,12,13]. These quantities are known to be the basic variables of the classical and quantum information theory of physical systems [6,14,15], and they characterize the uncertainty measures of these systems in a much better way than the Heisenberg-like measures since they do not depend on a specific point of the domain of the density and do not give a large weight to the tails of the distribution (see, e.g., [16]), which is only true for some particular distributions such as those that fall off exponentially. Moreover, they have allowed for gaining a much deeper knowledge of many scientific and technological phenomena ranging from the uncertainty principle of quantum physics [17,18,19,20,21,22], quantum entanglement and Bose–Einstein condensates [23,24,25,26,27], statistical mechanics [28,29,30,31,32,33,34,35], free and confined quantum systems [36,37,38,39,40], and Aharonov–Bohm rings in external fields [41,42] to biology and medicine [43,44,45].

The Rényi entropies of general quantum systems, which are power-like functionals of the single-particle probability density, cannot be exactly determined from first principles because the associated Schrödinger equation in both position and momentum spaces is unsolvable. Since the analytical properties of the Rényi quantities have been widely examined (see, e.g., [17,46,47,48]) and reviewed (see, e.g., [18,19,31]), a number of rigorous bounds have been found for general and central-potential quantum systems as it is described below, beginning with the entropic uncertainty relation.

Even though the Schrödinger equation is solvable, which happens for a small bunch of quantum-mechanical potentials such as the infinite-well potential [37] and the rigid rotator [38] and in the oscillator (harmonic-like) and hydrogenic (Coulomb-like) systems [49,50,51,52,53], the exact determination of the Rényi entropies for their stationary states is a formidable task except for the first few lowest-lying energetic states of some specific systems and for a few one-dimensional exponential densities (see, e.g., [54]). In the last few years, however, it has been recently solved for the multidimensional oscillator and hydrogenic sytems as it is discussed in this work. This has been possible because the Rényi quantities of these systems are integral functionals of the known hypergeometric orthogonal polynomials (see, e.g., [55,56,57]), which control the state’s wavefunctions. They can be algorithmically calculated in terms of the potential strength, the space dimensionality, and the state’s hyperquantum numbers [58,59] by using linearization methods of orthogonal polynomials [60,61].

Nevertheless, when applied to the hardest extreme high-energy (Rydberg) [62,63,64] and high-dimensional (quasiclassical) states, the resulting general expressions are computationally demanding because the corresponding Rényi integral functional kernels are highly oscillatory when the radial hyperquantum number and the space dimensionality become large. For such special states, it is much more convenient and physically transparent to use the degree and parameter asymptotics [65,66,67,68,69] of the Laguerre and Gegenbauer polynomials, which control the radial and angular parts of corresponding state’s wavefunctions. Indeed, these asymptotical techniques have allowed us to express the Rényi entropies of the high-lying excited Rydberg states [70,71,72,73,74,75] and high-dimensional states [70,76,77] of oscillator and hydrogenic systems in a simple, compact way. Due to their extraordinary properties (large dipole polarizability, long-range dipolar interactions,....), we will focus on the Rydberg states because not only are they a fertile laboratory to investigate the order-to-chaos transitions through the applications of electric fields [63,78] and to explore the strongly interacting systems [79], but they are also one of the most promising neutral-atom platforms in various quantum-information tasks (see, e.g., [80,81,82,83,84]).

The structure of the paper is as follows: In Section 2, we fix the notation used and give the physical solutions in both position and momentum space for the Schrödinger equation of the multidimensional oscillator-like and hydrogenic systems in terms of the hyperspherical quantum numbers and the potential strength. In Section 3, we show the lower and upper bounds for the Rényi entropies of general and central-potential quantum systems, as well as the associated entropic uncertainty relations. In Section 4, we show how to determine the Rényi entropies for arbitrary states of the multidimensional oscillator-like systems in both spherical and cartesian coordinates. In Section 5, the Rényi entropies for general states of the multidimensional hydrogenic systems are analytically found and discussed. In Section 6, powerful asymptotical techniques are used to obtain the Rényi entropies for the highly-excited Rydberg states of the multidimensional oscillator and hydrogenic states. Finally, some conclusions and open problems are given.

## 2. The *D*-Dimensional Oscillator and Hydrogenic Eigenvalue Problems

The time-independent non-relativistic equation of a *D*-dimensional (D⩾1) single-particle system subject to the quantum-mechanical potential $V_{D}\left(r\right)$ is given by the Schrödinger equation
(3)$$\left(-\frac{1}{2}{\vec{\nabla }}_{D}^{2}+V_{D}\left(r\right)\right)\Psi \left(r\right)=E\Psi \left(r\right),$$
where ${\vec{\nabla }}_{D}$ denotes the *D*-dimensional gradient operator, and the position vector $r=\left(x_{1},\ldots ,x_{D}\right)=\left(r,\theta_{1},\theta_{2},\ldots ,\theta_{D-1}\right)$ in Cartesian and hyperspherical units, respectively. Moreover, $r\equiv \left|r\right|=\sqrt{\sum_{i=1}^{D}x_{i}^{2}}\in \left[0,\;+\infty \right)$ and $x_{i}=r\left(\prod_{k=1}^{i-1}\sin\theta_{k}\right)\cos\theta_{i}$ for 1≤i≤D and with $\theta_{i}\in \left[0,\pi \right),i<D-1$, $\theta_{D-1}\equiv \phi \in \left[0,2\pi \right)$. Atomic units (i.e., $\hbar =m_{e}=e=1$) are used throughout the paper.

In this section, we give the wavefunctions (i.e., the eigenfunctions Ψ and the energetic eigenvalue *E*) and the associated probability densities for the bound stationary states of the *D*-dimensional isotropic harmonic oscillator and hydrogenic systems in both position and momentum spaces.

### 2.1. The *D*-Dimensional Oscillator Eigenvalue Problem

This problem corresponds to an isotropic harmonic oscillator [85] described by the potential $V_{D}^{\left(O\right)}\left(r\right)=\frac{1}{2}\omega^{2}r^{2}$, where ω is the oscillator strength. In Cartesian units, the wavefunctions are described (see, e.g., [58]) by the Cartesian quantum numbers $\left\{n_{i}\right\}\equiv \left(n_{1},n_{2},...,n_{D}\right)$, since the energetic eigenvalues
(4)$$E_{\left\{n_{i}\right\}}^{\left(O\right)}=\left(N+\frac{D}{2}\right)\omega ,\;\mathrm{with}\;N=\sum_{i=1}^{D}n_{i}\;;\;n_{i}=0,1,2,\ldots$$
and the associated eigenfunctions
(5)$$\Psi_{\left\{n_{i}\right\}}^{\left(O\right)}\left(r\right)=N\;e^{-\frac{1}{2}\alpha^{\prime }\left(x_{1}^{2}+\ldots +x_{D}^{2}\right)}H_{n_{1}}\left(\sqrt{\alpha^{\prime }}\;x_{1}\right)\cdots H_{n_{D}}\left(\sqrt{\alpha^{\prime }}\;x_{D}\right),$$
where $\alpha^{\prime }=\omega^{\frac{1}{4}}$, $H_{n_{i}}\left(x\right)$ denotes the Hermite polynomial of degree $n_{i}$ orthogonal [55,56,57]) with respect the weight function $\omega \left(x\right)=e^{-x^{2}}$ in (−∞,∞), and N denotes the normalization constant
$$N=\frac{1}{\sqrt{2^{N}n_{1}!n_{2}!\cdots n_{D}!}}{\left(\frac{\alpha^{\prime }}{\pi }\right)}^{D/4}.$$
The probability density of the *D*-dimensional isotropic harmonic oscillator is given by the modulus squared of the corresponding eigenfunctions, obtaining the expressions
(6)$$\rho_{\left\{n_{i}\right\}}^{\left(O\right)}\left(r\right)={\left|\psi_{\left\{n_{i}\right\}}\left(r\right)\right|}^{2}=N^{2}\;e^{-\alpha^{\prime }\left(x_{1}^{2}+x_{2}^{2}+\ldots +x_{D}^{2}\right)}H_{n_{1}}^{2}\left(\sqrt{\alpha^{\prime }}\;x_{1}\right)\cdots H_{n_{D}}^{2}\left(\sqrt{\alpha^{\prime }}\;x_{D}\right),$$
in position space. Working similarly in momentum space, one has
(7)$$\begin{array}{cc} \gamma_{\left\{n_{i}\right\}}^{\left(O\right)}\left(p\right) & ={\tilde{N}}^{2}e^{-\frac{1}{\alpha^{\prime }}\left(p_{1}^{2}+p_{2}^{2}+\ldots +p_{D}^{2}\right)}H_{n_{1}}^{2}\left(\frac{p_{1}}{\sqrt{\alpha^{\prime }}}\right)\cdots H_{n_{D}}^{2}\left(\frac{p_{D}}{\sqrt{\alpha^{\prime }}}\right)={\alpha \prime }^{-D}\rho_{N}\left(\frac{p}{\alpha^{\prime }}\right) \end{array}$$
with the normalization constant
$$\tilde{N}=\frac{1}{\sqrt{2^{N}n_{1}!n_{2}!\cdots n_{D}!}}{\left(\frac{1}{\pi \alpha^{\prime }}\right)}^{D/4}.$$

In hyperspherical units, the oscillator wavefunctions ($E_{n,l}^{\left(O\right)},\Psi_{n_{r},l,\left\{\mu \right\}}^{\left(O\right)}\left(r\right)$) are equivalently described [85] by the *D* hyperquantum integer numbers $\left(n_{r},l,\left\{\mu \right\}\right)\equiv \left(n_{r},\mu_{1},\mu_{2},\ldots ,\mu_{D-1}\right)$ with the values $n_{r}=0,1,2,\ldots ,\;l=0,1,2,\ldots ,$ and $l\geq \mu_{1}\geq \mu_{2}\geq \ldots \geq |\mu_{D-1}\left|\equiv |m\right|$, so that the energies
(8)$$E_{n,l}^{\left(O\right)}=\left(\eta +\frac{3}{2}\right)\;\omega =\left(2n_{r}+l+\frac{D}{2}\right)\omega ,\;\mathrm{with}\;\eta =n+\frac{D-3}{2},\;n=2n_{r}+l$$
and the associated position eigenfunctions
(9)$$\begin{array}{ccc} \Psi_{n_{r},l,\left\{\mu \right\}}^{\left(O\right)}\left(r\right) & = & {\left[\frac{2n_{r}!\;\omega^{l+\frac{D}{2}}}{\Gamma (n_{r}+l+\frac{D}{2})}\right]}^{\frac{1}{2}}r^{l}e^{-\frac{\omega \;r^{2}}{2}}L_{n_{r}}^{\left(l+D/2-1\right)}\left(\omega \;r^{2}\right)\;\times Y_{l,\{\mu \}}\left(\Omega_{D}\right), \end{array}$$
where the angular part is given by the hyperspherical harmonics [85,86,87]
(10)$$Y_{l,\left\{\mu \right\}}\left(\Omega \right)=\frac{1}{\sqrt{2\pi }}e^{im\theta_{D-1}}\prod_{j=1}^{D-2}{\tilde{C}}_{\mu_{j}-\mu_{j+1}}^{\left(\alpha_{j}+\mu_{j+1}\right)}\left(\cos\theta_{j}\right){\left(\sin\theta_{j}\right)}^{\mu_{j+1}},$$
with $2\alpha_{j}=D-j-1$. The symbol ${\tilde{C}}_{n}^{\left(\lambda \right)}\left(x\right)$, $\lambda >-\frac{1}{2}$, denotes the Gegenbauer polynomial orthonormal [55,57]) with respect to the weight function $\omega_{\lambda }^{\prime }\left(x\right)={\left(1-x^{2}\right)}^{\lambda -\frac{1}{2}},$ so that it fulfills
(11)$$\int_{-1}^{1}{\tilde{C}}_{n}^{\left(\lambda \right)}\left(x\right){\tilde{C}}_{m}^{\left(\lambda \right)}\left(x\right)\;\omega_{\lambda }\left(x\right)dx=\delta_{mn},$$
Note that the radial part is controlled by the Laguerre polynomials $L_{n}^{\left(\alpha \right)}\left(x\right)$ orthogonal [55,56,57] with respect to the weight function $\omega_{\alpha }\left(x\right)=x^{\alpha }e^{-x},\;\alpha =l+\frac{D}{2}-1,$ on the interval $\left[0,\infty \right)$. Then, the position probability density of the *D*-dimensional isotropic harmonic oscillator in hyperspherical units is given by
(12)$$\begin{array}{ccc} \rho_{n_{r},l,\left\{\mu \right\}}^{\left(O\right)}\left(r\right) & = & \frac{2n_{r}!\;\omega^{l+\frac{D}{2}}}{\Gamma (n_{r}+l+\frac{D}{2})}r^{2l}e^{-\omega \;r^{2}}{\left[L_{n_{r}}^{\left(l+D/2-1\right)}\left(\omega \;r^{2}\right)\right]}^{2}\times {\left|Y_{l,\{\mu \}}\left(\Omega_{D-1}\right)\right|}^{2}\\ & \equiv & \rho_{n_{r},l}^{\left(O\right)}\left(r\right)\times \rho_{l,\{\mu \}}\left(\Omega_{D-1}\right) \end{array}$$
where $\tilde{r}=\omega \;r^{2}$. Moreover, since the momentum wavefunctions ${\hat{\Psi }}_{n_{r},l,\left\{\mu \right\}}\left(p\right)$ are the Fourier transform of the position ones $\Psi_{n_{r},l,\left\{\mu \right\}}\left(r\right)$, we have that the following expressions
(13)$$\begin{array}{ccc} {\hat{\Psi }}_{n_{r},l,\left\{\mu \right\}}^{\left(O\right)}\left(p\right) & = & {\left[\frac{2n_{r}!\;\omega^{-l-\frac{D}{2}}}{\Gamma (n_{r}+l+\frac{D}{2})}\right]}^{\frac{1}{2}}p^{l}e^{-\frac{p^{2}}{2\omega }}L_{n_{r}}^{\left(l+D/2-1\right)}\left(\frac{p^{2}}{\omega }\right)\;\times Y_{l,\{\mu \}}\left(\Omega_{D-1}\right), \end{array}$$
and
(14)$$\begin{array}{ccc} \gamma_{n_{r},l,\left\{\mu \right\}}^{\left(O\right)}\left(p\right) & = & \frac{2n!\;\omega^{-l-\frac{D}{2}}}{\Gamma (n+l+\frac{D}{2})}p^{2l}e^{-\frac{p^{2}}{\omega }}{\left[L_{n}^{\left(l+D/2-1\right)}\left(\frac{p^{2}}{\omega }\right)\right]}^{2}\times {\left|Y_{l,\{\mu \}}\left(\Omega_{D-1}\right)\right|}^{2}\\ & = & \frac{1}{\omega^{D}}\;\rho_{n_{r},l,\left\{\mu \right\}}^{\left(O\right)}\left(\frac{p}{\omega }\right), \end{array}$$
give the momentum wavefunctions and the associated probability densities of the *D*-dimensional isotropic harmonic oscillator in hyperspherical units, respectively.

Finally, note that the wavefunctions are normalized to unity so that $\int {\left|\Psi_{n_{r},l,\left\{\mu \right\}}^{\left(O\right)}\left(r\right)\right|}^{2}dr=\int {\left|\Psi_{n_{r},l,\left\{\mu \right\}}^{\left(O\right)}\left(p\right)\right|}^{2}dp=1$, where the *D*-dimensional volume element is $dr=r^{D-1}dr\;d\Omega_{D-1}$ and $dp=r^{D-1}dr\;d\Omega_{D-1}$, respectively, with the generalized solid angle element
$$d\Omega_{D-1}=\left(\prod_{j=1}^{D-2}{\left(\sin\theta_{j}\right)}^{2\alpha_{j}}d\theta_{j}\right)d\theta_{D-1},$$
and we have also used the normalization of the hyperspherical harmonics given by $\int |Y_{l,\{\mu \}}\left(\Omega_{D-1}\right){|}^{2}d\Omega_{D-1}=1$.

### 2.2. The *D*-Dimensional Hydrogenic Eigenvalue Problem

This problem corresponds to a particle moving under a *D*-dimensional (D>1) central potential of the Coulomb form $V_{D}^{\left(H\right)}\left(r\right)=-\frac{Z}{r}$, where *Z* is the nuclear charge. In hyperspherical units, the wavefunctions of the stationary bound hydrogenic states are known [85] to be described by the hyperquantum numbers (n,l,{μ}). This is because the energetic eigenvalues $E_{\eta }^{\left(H\right)}=-\frac{Z^{2}}{\eta^{2}}$ (with the grand quantum number $\eta =n+\frac{D-3}{2}$, and n=1,2,…) and the associated eigenfunctions are given by
(15)$$\Psi_{n,l,\{\mu \}}^{\left(H\right)}\left(r\right)=R_{n,l}^{\left(H\right)}\left(r\right)\times Y_{l,\{\mu \}}\left(\Omega_{D-1}\right),$$
where the radial part is given as
(16)$$R_{n,l}^{\left(H\right)}\left(r\right)={\left(\frac{{\lambda \prime }^{-D}}{2\eta }\right)}^{\frac{1}{2}}{\left[\frac{\omega_{2L+1}\left(\tilde{r}\right)}{{\tilde{r}}^{D-2}}\right]}^{\frac{1}{2}}{\tilde{L}}_{\eta -L-1}^{\left(2L+1\right)}\left(\tilde{r}\right)$$
where the grand orbital angular momentum quantum number $L=l+\frac{D-3}{2}$, 2L+1=2l+D−2, the parameter $\lambda^{\prime }=\frac{\eta }{2Z}$, and $\tilde{r}=\frac{r}{\lambda^{\prime }}$. The symbols $L_{m}^{\left(\alpha \right)}\left(x\right)$ and ${\tilde{L}}_{m}^{\left(\alpha \right)}\left(x\right)$ denote the orthogonal and orthonormal Laguerre polynomials with respect to the weight $\omega_{\alpha }\left(x\right)=x^{\alpha }e^{-x}$ on the interval $\left[0,\infty \right)$, respectively, so that
(17)$${\tilde{L}}_{m}^{\left(\alpha \right)}\left(x\right)={\left[\frac{m!}{\Gamma (m+\alpha +1)}\right]}^{\frac{1}{2}}L_{m}^{\left(\alpha \right)}\left(x\right).$$

Then, the associated probability density is given by
(18)$$\begin{array}{ccc} \rho_{n,l,\{\mu \}}^{\left(H\right)}\left(r\right) & = & \frac{{\lambda \prime }^{-D}}{2\eta }\frac{\omega_{2L+1}\left(\tilde{r}\right)}{{\tilde{r}}^{D-2}}{\left[{\tilde{L}}_{\eta -L-1}^{\left(2L+1\right)}\left(\tilde{r}\right)\right]}^{2}\times {\left|Y_{l,\{\mu \}}\left(\Omega_{D-1}\right)\right|}^{2}\\ & \equiv & \rho_{n,l}^{\left(H\right)}\left(r\right)\times \rho_{l,\{\mu \}}\left(\Omega_{D-1}\right) \end{array}$$
in position space, where the hyperquantum numbers $\left(n,l,\left\{\mu \right\}\right)=\left(n,l\equiv \mu_{1},\ldots ,\mu_{D-1}\right)$ corresponding to the polar hyperspherical coordinates $r=(r,\theta_{1},\theta_{2},\ldots ,\theta_{D-1})$, have the values {$l=0,1,2,\ldots ,n-1;\;l\geq \mu_{2}\geq \ldots \geq \mu_{D-1}\equiv \left|m\right|\geq 0$}.

Working similarly in momentum space, one has the following expression [85,88]
(19)$$\begin{array}{ccc} \gamma_{n,l,\{\mu \}}^{\left(H\right)}\left(p\right) & = & {\left(\frac{\eta }{Z}\right)}^{D}{\left(1+y\right)}^{3}{\left(\frac{1+y}{1-y}\right)}^{\frac{D-2}{2}}\omega_{L+1}^{*}\left(y\right){\left[{\tilde{C}}_{\eta -L-1}^{\left(L+1\right)}\left(y\right)\right]}^{2}\times {\left[Y_{l,\{\mu \}}\left(\Omega_{D-1}\right)\right]}^{2}\\ & = & K_{n,l}^{2}\frac{{\left(\eta \tilde{p}\right)}^{2l}}{{\left(1+\eta^{2}{\tilde{p}}^{2}\right)}^{2L+4}}{\left[C_{\eta -L-1}^{\left(L+1\right)}\left(\frac{1-\eta^{2}{\tilde{p}}^{2}}{1+\eta^{2}{\tilde{p}}^{2}}\right)\right]}^{2}{\left|Y_{l,\{\mu \}}\left(\Omega_{D-1}\right)\right|}^{2}\\ & \equiv & \gamma_{n,l}^{\left(H\right)}\left(r\right)\times \rho_{l,\{\mu \}}\left(\Omega_{D-1}\right) \end{array}$$
for the probability density of these systems in the *D*-dimensional momentum space with the notation $\tilde{p}=\frac{p}{Z}$, the variable $y\equiv \frac{1-\eta^{2}{\tilde{p}}^{2}}{1+\eta^{2}{\tilde{p}}^{2}}$, the constant
(20)$$K_{n,l}=Z^{-\frac{D}{2}}2^{2L+3}{\left[\frac{(\eta -L-1)!}{2\pi (\eta +L)!}\right]}^{\frac{1}{2}}\Gamma \left(L+1\right)\eta^{\frac{D+1}{2}}.$$
and the symbol $C_{n}^{\left(\lambda \right)}\left(x\right)$, $\lambda >-\frac{1}{2}$ denotes the Gegenbauer polynomial orthogonal [57] with respect to the weight function $\omega_{\lambda }^{\prime }\left(x\right)={\left(1-x^{2}\right)}^{\lambda -\frac{1}{2}}.$

## 3. Rényi Entropies of General and Central-Potential Quantum Systems: Lower and Upper Bounds

In this section, we first describe the lower and upper bounds [48,89] on the Rényi entropies (Equation 1) and (Equation 2) of general multidimensional quantum systems in position and momentum spaces, and the corresponding entropic uncertainty relations [17,20,21,90,91]. Then, we show the improvement of these properties for central potentials, and we point out some open problems [89,92,93].

The most relevant property of the Rényi entropies (Equation 1) and (Equation 2) of general *D*-dimensional quantum systems is the entropic uncertainty relation
(21)$$R_{q}\left[\rho_{\left\{n_{i}\right\}}\right]+R_{q^{*}}\left[\gamma_{\left\{n_{i}\right\}}\right]\geq D\ln\left(\pi q^{\frac{1}{2q-2}}{q^{*}}^{\frac{1}{2q^{*}-2}}\right)$$
which is saturated by the Gaussian distributions. This relation was proved by Zozor, Portesi and Vignat [21] for arbitrary indices, extending the one-dimensional relation previously found by Bialynicki-Birula [20] and Zozor and Vignat [90] for conjugated indices (i.e., when $\frac{1}{q}+\frac{1}{q^{*}}=2$); see [17,91] for further details. See [94,95] for other related inequality relations.

Moreover, the Rényi entropies fulfill the monotonicity relations given by
(22)$$R_{p}\left[\rho \right]\geq R_{q}\left[\rho \right],\;\mathrm{if}\;p\leq q;\;\mathrm{and}\;\frac{p-1}{p}R_{p}\left[\rho \right]\geq \frac{q-1}{q}R_{q}\left[\rho \right],\;\mathrm{if}\;p\geq q>1,$$
which, among many other consequences [30,48], allows one to lowerbound all the Rényi entropies by means of the second-order entropy as $R_{q}\left[\rho \right]\geq \frac{1}{2}R_{2}\left[\rho \right],\;\mathrm{for}\;q>0$. See [96] for further related inequalities. In addition, the position Rényi entropies (Equation 1) can be upperbounded [93] in terms of the Heisnberg measure $\langle r^{2}\rangle$ as
(23)$$R_{q}\left[\rho \right]\leq B_{D}\left(q\right)+\frac{D}{2}\ln\left(\frac{\langle r^{2}\rangle }{D}\right),$$
with
(24)$$B_{D}\left(q\right)=\left\{\begin{array}{ccc} \frac{D}{2}\ln\left(\frac{\pi ((2+D)q-D)}{q-1}\right)+\frac{1}{q-1}\ln\left(\frac{(2+D)q-D}{2q}\right)+\ln\left(\frac{\Gamma \left(\frac{q}{q-1}\right)}{\Gamma \left(\frac{(2+D)q-D}{2(q-1)}\right)}\right) & \mathrm{if} & q>1,\;\\ \frac{D}{2}\ln\left(\frac{\pi ((2+D)q-D)}{1-q}\right)-\frac{q}{1-q}\ln\left(\frac{(2+D)q-D}{2q}\right)-\ln\left(\frac{\Gamma \left(\frac{q}{1-q}\right)}{\Gamma \left(\frac{(2+D)q-D}{2(1-q)}\right)}\right) & \mathrm{if} & q\in \left(\frac{D}{D+2},1\right),\; \end{array}\right.$$

A similar upper bound can be obtained for the momentum Rényi entropies (Equation 2), as given by
(25)$$R_{q^{*}}\left[\gamma \right]\leq B_{D}\left(q^{*}\right)+\frac{D}{2}\ln\left(\frac{\langle p^{2}\rangle }{D}\right),$$
The combination of the upper bounds (Equation 23) and (Equation 25) gives rise to the following inequality between the position–momentum Rényi-entropy sum and the position–momentum Heisenberg uncertainty $\langle r^{2}\rangle \langle p^{2}\rangle$:(26)$$R_{q}\left[\rho \right]+R_{q^{*}}\left[\gamma \right]\leq B_{D}\left(q\right)+B_{D}\left(q^{*}\right)-D\ln D+\frac{D}{2}\left(\langle r^{2}\rangle \langle p^{2}\rangle \right),$$
which extends the corresponding three-dimensional result [89] to *D* dimensions, emphasizes the uncertainty character of the position–momentum Rényi-entropy sum, and complements the entropic uncertainty relation (Equation 21).

The expression (Equation 23) has been variationally extended [97,98] in both conjugated spaces by using the Heisenberg measures $\langle r^{k}\rangle$ and $\langle p^{k}\rangle$, respectively, with integer $k<\frac{D}{q}\left(q-1\right)$. The resulting general expressions generalize (see also [99]) previous bounds obtained in the one-dimensional [100,101] and three-dimensional [102] cases used in various contexts, ranging from financial and quantum technologies.

Let us now consider the improvement of all these previous properties for quantum systems subject to a central potential, whose Schrödinger equation is given by Equation (Equation 3). In this case, the position and momentum probability densities of the *D*-dimensional stationary state (n,l,{μ}) are given by the modulus squared of the two corresponding eigenfunctions as
(27)$$\begin{array}{ccc} \rho_{n,l,\{\mu \}}\left(r\right) & = & {\left|\Psi_{n,l,\{\mu \}}\left(r\right)\right|}^{2}=|R_{nl}{\left(r\right)|}^{2}\times {\left|Y_{l,\{\mu \}}\left(\Omega_{D-1}\right)\right|}^{2} \end{array}$$
(28)$$\begin{array}{ccc} & = & r^{1D}\;|u_{nl}{\left(r\right)|}^{2}\times {\left|Y_{l,\{\mu \}}\left(\Omega_{D-1}\right)\right|}^{2} \end{array}$$
in position space, and
(29)$$\begin{array}{ccc} \gamma_{n,l,\{\mu \}}\left(p\right) & = & {\left|{\tilde{\Psi }}_{n,l,\{\mu \}}\left(p\right)\right|}^{2}={\left|M_{n,l}\left(p\right)\right|}^{2}\times {\left|Y_{l,\{\mu \}}\left({\hat{\Omega }}_{D-1}\right)\right|}^{2} \end{array}$$
(30)$$\begin{array}{ccc} & = & p^{1D}\;|{\tilde{u}}_{nl}{\left(p\right)|}^{2}\times {\left|Y_{l,\{\mu \}}\left(\Omega_{D-1}\right)\right|}^{2} \end{array}$$
in momentum space, where the reduced radial momentum eigenfunction
(31)$${\tilde{u}}_{nl}\left(p\right)={\left(-i\right)}^{l}\int_{0}^{\infty }\sqrt{r\;p}\;\;J_{l+D/2-1}\left(r\;p\right)\;u\left(r\right)dr,$$
is the Hankel transform of the reduced radial position eigenfunction $u_{nl}\left(r\right)$.

To improve the upper bounds (Equation 23) and (Equation 25) for central potentials in terms of the expectation values $\langle r^{2}\rangle$ and $\langle p^{2}\rangle$, respectively, we use the Rényi maximization procedure of Costa et al. [103] to find [93] the following sharp inequality:(32)$$R_{q}\left[\rho \right]\leq B_{D}\left(q\right)+\frac{D}{2}\ln\left(\frac{\langle r^{2}\rangle }{D}\right)+L\left(\Omega_{D-1}\right),$$
where the central-potential effects are contained in the quantity $L(\Omega_{D-1})$, which represents the loss of entropy due to the angular part of the state’s wavefunction, is given by
(33)$$L\left(\Omega_{D-1}\right)=\frac{1}{2}\sum_{k=1}^{D-2}\left(\left(D-k\right)\ln\langle \sin^{2}\theta_{k}\rangle +\ln\langle \cos^{2}\theta_{k}\rangle \right)-\ln2+\frac{D}{2}\ln D,$$
and
(34)$$\langle \cos^{2}\theta_{k}\rangle =\frac{2\mu_{k}\left(\mu_{k}+D-k-1\right)-2\mu_{k+1}\left(\mu_{k+1}+D-k-2\right)+D-k-3}{4\mu_{k}\left(\mu_{k}+D-k-1\right)+\left(D-k+1\right)\left(D-k-3\right)}.$$
and, of course, $\langle \sin^{2}\theta_{k}\rangle =1-\langle \cos^{2}\theta_{k}\rangle$. The similar upper bound on the momentum Rényi entropy of central potentials is found to be
(35)$$R_{q^{*}}\left[\gamma \right]\leq B_{D}\left(q^{*}\right)+\frac{D}{2}\ln\left(\frac{\langle p^{2}\rangle }{D}\right)+L\left(\Omega_{D-1}\right)$$
Thus, the position–momentum Rényi entropy sum and the position–momentum Heisenberg uncertainty $\langle r^{2}\rangle \langle p^{2}\rangle$ are related as
(36)$$R_{q}\left[\rho \right]+R_{q^{*}}\left[\gamma \right]\leq B_{D}\left(q\right)+B_{D}\left(q^{*}\right)-D\ln D+\frac{D}{2}\left(\langle r^{2}\rangle \langle p^{2}\rangle \right)+2\;L\left(\Omega_{D-1}\right)$$
for central potentials with dimensionality D≥3, which considerably improves the previous inequality (Equation 26) valid for general quantum systems.

Finally, let us mention that the improvement of the general entropic uncertainty relation (Equation 21) for central potentials is an open problem yet. However, an heuristic method [92] has recently allowed us to find the following uncertainty relation
(37)$$R_{q}\left[\rho_{n,l,\{\mu \}}\right]+R_{q*}\left[\gamma_{n,l,\{\mu \}}\right]\geq \frac{2q\ln A(2q)}{q-1}+\frac{2q^{*}\ln A\left(2q^{*}\right)}{q^{*}-1}+R_{q}\left[Y_{l,\{\mu \}}\right]+R_{q*}\left[Y_{l,\{\mu \}}\right],$$
where the constant
$$A\left(q\right)=2^{\frac{2D}{2q}}\frac{q^{\frac{1}{2}\left(\frac{1}{2}+l+\frac{D}{2}-1+\frac{D-1}{2}\frac{2-q}{q}+\frac{1}{q}\right)}}{\Gamma {\left(\left(l+\frac{D}{2}-1+\frac{D-1}{2}\frac{2-q}{q}+\frac{1}{2}\right)\frac{q}{2}+\frac{1}{2}\right)}^{\frac{1}{q}}},$$
and the angular Rényi entropies are given by
(38)$$R_{q}\left[Y_{l,\{\mu \}}\right]:=\frac{1}{1-q}\ln\Lambda_{q}\left[Y_{l,\{\mu \}}\right]$$
with the integral functionals of the hyperspherical harmonics [77]
(39)$$\begin{array}{ccc} \Lambda_{q}\left[Y_{l,\{\mu \}}\right] & = & \int_{S_{D-1}}{\left|Y_{l,\{\mu \}}\left(\Omega_{D-1}\right)\right|}^{2q}\;d\Omega_{D-1}\\ & = & 2\pi N_{l,\{\mu \}}^{2q}\prod_{j=1}^{D-2}\int_{0}^{\pi }{\left[C_{\mu_{j}-\mu_{j+1}}^{\left(\alpha_{j}+\mu_{j+1}\right)}\left(\cos\theta_{j}\right)\right]}^{2q}{\left(\sin\theta_{j}\right)}^{2q\mu_{j+1}+2\alpha_{j}}\;d\theta_{j}, \end{array}$$
and the normalization constant $N_{l,\{\mu \}}$ is given by
(40)$$N_{l,\{\mu \}}^{2}=\frac{1}{2\pi }\prod_{j=1}^{D-2}\frac{\left(\alpha_{j}+\mu_{j}\right)\left(\mu_{j}-\mu_{j+1}\right)!{\left[\Gamma \left(\alpha_{j}+\mu_{j+1}\right)\right]}^{2}}{\pi \;2^{1-2\alpha_{j}-2\mu_{j+1}}\Gamma \left(2\alpha_{j}+\mu_{j}+\mu_{j+1}\right)}.$$

This heuristic uncertainty relation (Equation 37) is not valid for all central potentials [92]. However, it has been numerically shown to be fulfilled by various large classes of qualitatively different central potentials such as, e.g., the oscillator and hydrogenic-like potentials. Note that, according Equations (Equation 10) and (Equation 59)–(Equation 40), the hyperspherical harmonics $Y_{l,\{\mu \}}\left(\Omega_{D-1}\right)$ and consequently the angular Rényi entropies $R_{q}\left[Y_{l,\{\mu \}}\right]$ do not depend on the principal hyperquantum number *n*, but they do depend on the angular hyperquantum numbers (l,{μ}) and the dimensionality *D*. Moreover, the integral functionals involved in (39) are the Rényi-like functionals of the Gegenbauer polynomials, which are under control since they can be analytically calculated by two recent methodologies: one based on the Srivastava’s linearization method [60,61] and another one based on the combinatorial Bell polynomials [104].

## 4. The Rényi Entropies of Multidimensional Harmonic Systems

In this section, we examine the position and momentum Rényi entropies, $R_{q}\left[\rho_{n_{r},l,\left\{\mu \right\}}^{\left(O\right)}\right]$ and $R_{q}\left[\gamma_{n_{r},l,\left\{\mu \right\}}^{\left(O\right)}\right]$, respectively, for any *D*-dimensional harmonic state. Then, the corresponding position–momentum entropic uncertainty sums are explicitly shown, which allows for a quantitative discussion of quantum uncertainty much richer than the conventional Heisenberg-like uncertainty [20,21] and its extension [105]. We first realize that these quantities cannot be explicitly expressed by hyperspherical quantum numbers $\left(n_{r},l,\left\{\mu \right\}\right)$, basically because they naturally depend on some power-like integral functionals of the Laguerre and Gegenbauer orthogonal polynomials whose analytical evaluation is not yet known (see, e.g., [85,106]). This is because the position Rényi entropies for an arbitrary *D*-dimensional oscillator-like state, which is characterized by the probability density $\rho_{n_{r},l,\left\{\mu \right\}}^{\left(O\right)}\left(r\right)$, can be expressed according to Equations (Equation 1) and (Equation 12) as
(41)$$\begin{array}{ccc} R_{q}\left[\rho_{n_{r},l,\left\{\mu \right\}}^{\left(O\right)}\right] & = & \frac{1}{1-q}\log\int_{R_{D}}{\left[\rho_{n_{r},l,\left\{\mu \right\}}^{\left(O\right)}\left(r\right)\right]}^{q}\;dr \end{array}$$
(42)$$\begin{array}{ccc} & = & R_{q}\left[\rho_{n_{r},l}^{\left(O\right)}\right]+R_{q}\left[\rho_{l,\{\mu \}}^{\left(O\right)}\right], \end{array}$$
where the symbols $R_{q}\left[\rho_{n_{r},l}^{\left(O\right)}\right]$ and $R_{q}\left[\rho_{l,\{\mu \}}^{\left(O\right)}\right]$ denote the radial and angular Rényi entropies for the *D*-dimensional harmonic state, respectively. The angular quantities $R_{q}^{\left(O\right)}\left[\rho_{l,\{\mu \}}\right]=R_{q}\left[Y_{l,\{\mu \}}\right]$ are given by the above-mentioned expressions (Equation 38)–(Equation 40), which do not depend on the principal hyperquantum number $n_{r}$, but they do depend on the hyperangular numbers (l,{μ}) and the dimensionality *D*. In addition, the radial quantities are given [76] by
(43)$$\begin{array}{ccc} R_{q}\left[\rho_{n_{r},l}^{\left(O\right)}\right] & = & \frac{1}{1-q}\log\int_{0}^{\infty }{\left[\rho_{n_{r},l}^{\left(O\right)}\left(r\right)\right]}^{p}\;r^{D-1}dr\\ & = & \frac{1}{1-q}\log N\left(n_{r},l,D,q\right)-\frac{D}{2}\log\omega -\log2 \end{array}$$
where the the weighted $L_{q}$-norm $N(n_{r},l,D,q)$ for the orthogonal and orthonormal Laguerre polynomials are given by
(44)$$\begin{array}{ccc} N(n_{r},l,D,q) & = & {\left(\frac{n_{r}!}{\Gamma (\alpha +n_{r}+1)}\right)}^{q}\int_{0}^{\infty }r^{\alpha +lq-l}e^{-qr}{\left[L_{n_{r}}^{\left(\alpha \right)}\left(r\right)\right]}^{2q}\;dr\\ & = & \int_{0}^{\infty }{\left({\left[{\tilde{L}}_{n_{r}}^{\left(\alpha \right)}\left(x\right)\right]}^{2}\;w_{\alpha }\left(x\right)\right)}^{q}\;x^{\beta }\;dx\;,\;q>0\; \end{array}$$
respectively, with $\alpha =l+\frac{D}{2}-1\;\;l=0,1,2,\ldots ,\;q>0\;\;\mathrm{and}\;\;\beta =\left(1-q\right)\left(\alpha -l\right)=\left(q-1\right)\left(1D/2\right)$; these values guarantee the convergence of the integral functional because the condition $\beta +q\alpha =\frac{D}{2}+lq-1>-1$ is always satisfied for physically meaningful values of the parameters. Similarly, from Equation (Equation 14), one has
(45)$$R_{q}\left[\gamma_{n_{r},l,\left\{\mu \right\}}^{\left(O\right)}\right]=R_{q}\left[\rho_{n_{r},l,\left\{\mu \right\}}^{\left(O\right)}\right]+D\;\log\;\omega$$
for the momentum Rényi entropies $R_{q}\left[\gamma_{n_{r},l,\left\{\mu \right\}}^{\left(O\right)}\right]$ of any *D*-dimensional harmonic state.

Thus, from expressions (Equation 59)–(Equation 45), we realize that, in hyperspherical coordinates, the Rényi entropies of the *D*-dimensional harmonic systems are controlled by some power-like integral functionals of the Gegenbauer and Laguerre polynomials as given by Equations (Equation 39) and (Equation 44), respectively, which have not yet been explicitly determined. Nevertheless, they can be analytically found in an algorithmic way by means of some generalized multivariate hypergeometric functions of a Srivastava–Karlsson type [60,61,107] or the multivariable Bell polynomials used in Combinatorics [104]; see also [108,109]. This has been recently illustrated in detail for the second-order Rényi entropies (thus the disequilibrium) of the *D*-dimensional harmonic systems [70].

The resulting algorithmic expressions are somewhat highbrow for the highly excited Rydberg states because of the large values of their principal hyperquantum numbers. For such extreme states, a powerful method based on the strong asymptotics of Laguerre polynomials [65,110] is much more convenient because it gives simple, transparent, and compact analytical expressions for the Rényi entropies of the *D*-dimensional harmonic systems. This is shown below in Section 6.1.

Alternatively, we can calculate in Cartesian units [58] the oscillator Rényi entropies $R_{q}\left[\rho_{\left\{n_{i}\right\}}^{\left(O\right)}\right],q\neq 1,$ for a generic state characterized by the Cartesian quantum numbers $\left\{n_{i}\right\}\equiv \left(n_{1},n_{2},...,n_{D}\right)$. This quantity, according to Equations (Equation 1) and (Equation 6), is given by
(46)$$\begin{array}{cc} R_{q}\left[\rho_{\left\{n_{i}\right\}}^{\left(O\right)}\right] & =\frac{1}{1-q}\log\int_{-\infty }^{\infty }dx_{1}\ldots \int_{-\infty }^{\infty }dx_{D}\;\left[\rho_{\left\{n_{i}\right\}}^{\left(O\right)}\right]\left(r\right){]}^{q}\\ \; & =\frac{1}{1-q}\log\left(N^{2q}\left[\Pi_{i=1}^{D}\int_{-\infty }^{\infty }e^{-\alpha^{\prime }qx_{i}^{2}}{\left|H_{n_{i}}\left(\sqrt{\alpha^{\prime }}\;x_{i}\right)\right|}^{2q}\;dx_{i}\right]\right) \end{array}$$

Now, to solve these integral functionals, we use the Srivastava-like linearization relation for the (2q)-th power of the Hermite polynomials [60], obtaining the following expression for the Rényi entropy of order *q* for the oscillator-like state
(47)$$\begin{array}{ccc} R_{q}\left[\rho_{\left\{n_{i}\right\}}^{\left(O\right)}\right] & = & -\frac{D}{2}\log\left[\alpha^{\prime }\right]+K_{q}\;D+{\bar{K}}_{q}\;N_{O}+\frac{q}{q-1}\sum_{i=1}^{D}{\left(-1\right)}^{n_{i}}\log\left[{\left(\frac{n_{i}+1}{2}\right)}_{\frac{1}{2}}\right]\\ & + & \frac{1}{1-q}\sum_{i=1}^{D}\log\left[F_{q}\left(n_{i}\right)\right], \end{array}$$
where ${\left(z\right)}_{a}=\frac{\Gamma (z+a)}{\Gamma (z)}$ is the known Pochhammer’s symbol,
(48)$$K_{q}=\frac{\log[\pi^{q-\frac{1}{2}}\;q^{\frac{1}{2}}]}{q-1};\;{\bar{K}}_{q}=\frac{1}{1-q}\log\left[\frac{4^{q}\;\Gamma \left(\frac{1}{2}+q\right)}{\pi^{\frac{1}{2}}\;q^{q}}\right],$$
the symbol $F_{q}\left(n_{i}\right)$ denotes the following multivariate Lauricella function of type A [107]:(49)$$\begin{array}{c} F_{q}\left(n\right)=\sum_{j_{1},\ldots ,j_{2q}=0}^{\frac{n-\nu }{2}}\frac{{\left(q\nu +\frac{1}{2}\right)}_{j_{1}+\ldots j_{2q}}{\left(\frac{\nu -n}{2}\right)}_{j_{1}}\cdots {\left(\frac{\nu -n}{2}\right)}_{j_{2q}}}{{\left(\nu +\frac{1}{2}\right)}_{j_{1}}\cdots {\left(\nu +\frac{1}{2}\right)}_{j_{2q}}}\frac{{\left(\frac{1}{q}\right)}^{j_{1}}\cdots {\left(\frac{1}{q}\right)}^{j_{2q}}}{j_{1}!\cdots j_{2q}!}, \end{array}$$
and the notation $N_{O}=\sum_{i=1}^{D}\nu_{i}$ is used for the amount of odd numbers $n_{i}$, so that $N_{E}=D-N_{O}$ gives the number of the even ones.

Note that the general expression (Equation 47) allows for the analytical determination of the Rényi entropies (with positive integer values of *q*) of a generic *D*-dimensional oscillator-like state. In particular, for the ground state (i.e., $n_{i}=0,\;i=1,\cdots ,D$, so N=0), this general expression gives rise to the value
(50)$$R_{q}\left[\rho_{\left\{0\right\}}^{\left(O\right)}\right]=\frac{D}{2}\log\left[\frac{\pi \;q^{\frac{1}{q-1}}}{\alpha^{\prime }}\right],\;q>0,$$
as one can also obtain directly from Equation (Equation 46).

Moreover, since the position and momentum densities are mutually rescaled, we have the following expression
(51)$$\begin{array}{ccc} R_{\tilde{q}}\left[\gamma_{\left\{n_{i}\right\}}^{\left(O\right)}\right] & = & \frac{D}{2}\log\left[\alpha^{\prime }\right]+K_{\tilde{q}}\;D+{\bar{K}}_{\tilde{q}}\;N_{O}+\frac{\tilde{q}}{\tilde{q}-1}\sum_{i=1}^{D}{\left(-1\right)}^{n_{i}}\log\left[{\left(\frac{n_{i}+1}{2}\right)}_{\frac{1}{2}}\right]\\ & + & \frac{1}{1-\tilde{q}}\sum_{i=1}^{D}\log\left[F_{\tilde{q}}\left(n_{i}\right)\right], \end{array}$$
for the associated momentum Rényi entropy ($\tilde{q}\in N$). Although Equations (Equation 47) and (Equation 51) rigorously hold for q≠1 and q∈N only, it is reasonable to conjecture its general validity for any q>0 and q≠1.

Then, from Equations (Equation 47) and (Equation 51), we have that the general expression for the position–momentum Rényi entropic sum is
(52)$$\begin{array}{ccc} R_{q}\left[\rho_{\left\{n_{i}\right\}}^{\left(O\right)}\right]+R_{\tilde{q}}\left[\gamma_{\left\{n_{i}\right\}}^{\left(O\right)}\right] & = & \left(K_{q}+K_{\tilde{q}}\right)\;D+\left({\bar{K}}_{q}+{\bar{K}}_{\tilde{q}}\right)\;N_{O}\\ & + & \left(\frac{q}{q-1}+\frac{\tilde{q}}{\tilde{q}-1}\right)\sum_{i=1}^{D}{\left(-1\right)}^{n_{i}}\log\left[{\left(\frac{n_{i}+1}{2}\right)}_{\frac{1}{2}}\right]\\ & + & \frac{1}{1-q}\sum_{i=1}^{D}\log\left[F_{q}\left(n_{i}\right)\right]+\frac{1}{1-\tilde{q}}\sum_{i=1}^{D}\log\left[F_{\tilde{q}}\left(n_{i}\right)\right], \end{array}$$
which verifies the Rényi-entropy-based uncertainty relation of Zozor–Portesi–Vignat [21] when $\frac{1}{q}+\frac{1}{\tilde{q}}\geq 2$ for arbitrary quantum systems. In the conjugated case, $\tilde{q}=q^{*}$ such that $\frac{1}{q}+\frac{1}{q^{*}}=2$, and one obtains
(53)$$\begin{array}{ccc} R_{q}\left[\rho_{\left\{n_{i}\right\}}^{\left(O\right)}\right]+R_{q^{*}}\left[\gamma_{\left\{n_{i}\right\}}^{\left(O\right)}\right] & = & D\log\left(\pi q^{\frac{1}{2q-2}}{q^{*}}^{\frac{1}{2q^{*}-2}}\right)+\left({\bar{K}}_{q}+{\bar{K}}_{q^{*}}\right)\;N_{O}\\ & + & \frac{1}{1-q}\sum_{i=1}^{D}\log\left[F_{q}\left(n_{i}\right)\right]+\frac{1}{1-q^{*}}\sum_{i=1}^{D}\log\left[F_{q^{*}}\left(n_{i}\right)\right]. \end{array}$$
We highlight that the first term of the right side of this relation is the sharp bound of the general uncertainty relation (Equation 21) valid for stationary states of arbitrary quantum systems [20,21]. In addition, the positivity for the sum of the remaining three terms can be shown, as expected.

## 5. The Rényi Entropies of Multidimensional Hydrogenic Systems

In this section, we show the analytical expressions for both position and momentum Rényi entropies $R_{q}\left[\rho_{n,l,\{\mu \}}^{\left(H\right)}\right],R_{q}\left[\gamma_{n,l,\{\mu \}}^{\left(H\right)}\right]$ (with natural *p* other than unity) for all discrete stationary states of the *D*-dimensional hydrogenic system in an algorithmic way. They are are expressed by means of some multiparametric hypergeometric functions of Lauricella and Srivastava–Daoust types [60,61,107]. We start with Equations (Equation 1), (Equation 18) and (Equation 19) obtaining
(54)$$\begin{array}{ccc} R_{q}\left[\rho_{n,l,\{\mu \}}^{\left(H\right)}\right] & = & \frac{1}{1-q}\log\int_{R_{D}}{\left[\rho_{n,l,\{\mu \}}^{\left(H\right)}\left(r\right)\right]}^{q}\;dr \end{array}$$
(55)$$\begin{array}{ccc} & = & R_{q}\left[\rho_{n,l}^{\left(H\right)}\right]+R_{q}\left[Y_{l,\{\mu \}}\right] \end{array}$$
in position space and
(56)$$\begin{array}{ccc} R_{q}\left[\gamma_{n,l,\{\mu \}}^{\left(H\right)}\right] & = & \frac{1}{1-q}\log\int_{R_{D}}{\left[\gamma_{n,l,\{\mu \}}^{\left(H\right)}\left(p\right)\right]}^{q}\;dp \end{array}$$
(57)$$\begin{array}{ccc} & = & R_{q}\left[\gamma_{n,l}^{\left(H\right)}\right]+R_{q}\left[Y_{l,\{\mu \}}\right] \end{array}$$
in momentum space, where the angular Rényi entropies $R_{q}\left[Y_{l,\{\mu \}}\right]$ are given by the expressions (Equation 38)–(Equation 40), and the radial Rényi entropies $R_{q}\left[\rho_{n,l}^{\left(H\right)}\right]$ and $R_{q}\left[\gamma_{n,l}^{\left(H\right)}\right]$ are given as
(58)$$R_{q}\left[\rho_{n,l}^{\left(H\right)}\right]=\frac{1}{1-q}\ln\int_{0}^{\infty }{\left[\rho_{n,l}^{\left(H\right)}\right]}^{q}r^{D-1}\;dr$$
and
(59)$$R_{q}\left[\gamma_{n,l}^{\left(H\right)}\right]=\frac{1}{1-q}\ln\int_{0}^{\infty }{\left[\gamma_{n,l}^{\left(H\right)}\right]}^{q}p^{D-1}\;dp$$
for the *D*-dimensional hydrogenic state in the two conjugated spaces, respectively.

Let us now determine the position radial entropy $R_{q}\left[\rho_{n,l}^{\left(H\right)}\right]$ given by Equations (Equation 58) and (Equation 18), which is
(60)$$\begin{array}{ccc} R_{q}\left[\rho_{n,l}^{\left(H\right)}\right] & = & \frac{1}{1-q}\ln\left[{\left(\frac{\eta }{2Z}\right)}^{D(1-q)}{\left(\frac{\Gamma (n-l)}{2\eta \Gamma (n+l+D-2)}\right)}^{q}\right]\\ & + & \frac{1}{1-q}\ln q^{-D-2lq}\int_{0}^{\infty }x^{2lq+D-1}e^{-x}{\left[L_{n-l-1}^{\left(2l+D-2\right)}\left(\frac{x}{q}\right)\right]}^{2q}\;dx, \end{array}$$
Then, we use the linearization formula [59,60] of the (2q)th-power of the Laguerre polynomial $L_{n-l-1}^{\left(2l+D-2\right)}\left(\frac{x}{q}\right)$ obtaining after some algebraic manipulations the expression
(61)$$\begin{array}{ccc} R_{q}\left[\rho_{n,l}^{\left(H\right)}\right]=D\ln\left(\frac{\eta }{2Z}\right)+\frac{q}{1-q}\ln\left(\frac{{\left(\eta -L\right)}_{2L+1}}{2\eta }\right)\\ & & \;+\frac{1}{1-q}\ln F_{q}\left(D,\eta ,L\right)+\frac{1}{1-q}\ln A_{q}\left(D,L\right) \end{array}$$
with $A_{q}\left(D,L\right)\equiv \frac{\Gamma \left(D+2lq\right)}{q^{D+2lq}\Gamma {\left(2L+2\right)}^{2q}}$, and
(62)$$\begin{array}{ccc} F_{q}\left(D,n,l\right) & & \equiv F_{A}^{\left(2q\right)}\left(\begin{array}{cc} 2lq+D;\overset{2q}{\overset{︷}{-n+l+1,\ldots ,-n+l+1}} & \\ & ;\underset{2q}{\underset{︸}{\frac{1}{q},\ldots ,\frac{1}{q}}}\\ \underset{2q}{\underset{︸}{2l+D-1,\ldots ,2l+D-1}} & \end{array}\right),\;\;\; \end{array}$$
where the symbol $F_{A}^{\left(s\right)}\left(x_{1},\ldots ,x_{r}\right)$ denotes the Lauricella function of type A of *s* variables and 2s+1 parameters defined [107] as
(63)$$F_{A}^{\left(s\right)}\left(\begin{array}{cc} a;b_{1},\ldots ,b_{s} & \\ \; & ;x_{1},\ldots ,x_{s}\\ c_{1},\ldots ,c_{s} & \end{array}\right)=\sum_{j_{1},\ldots ,j_{s}=0}^{\infty }\frac{{\left(a\right)}_{j_{1}+\ldots +j_{s}}{\left(b_{1}\right)}_{j_{1}}\cdots {\left(b_{s}\right)}_{j_{s}}}{{\left(c_{1}\right)}_{j_{1}}\cdots {\left(c_{s}\right)}_{j_{s}}}\frac{x_{1}^{j_{1}}\cdots x_{s}^{j_{s}}}{j_{1}!\cdots j_{s}!}.$$
Let us emphasize that the function $F_{q}\left(D,n,l\right)$ is a finite sum because of the properties of the involved Pochhammer symbols with negative integer arguments. Moreover, for l=n−1, the function $F_{q}\left(D,n,l\right)$ in Equation (Equation 61) is equal to unity, so that the third term on the right side vanishes. Then, for the ground state n=1, we have the simple values
$$\begin{array}{ccc} R_{q}\left[\rho_{1,0}^{\left(H\right)}\right] & = & \Gamma \left(D\right)+D\ln\left[\frac{D-1}{4Z\;q^{\frac{1}{1-q}}}\right] \end{array}$$

Working similarly with Equations (Equation 59), (Equation 19), and (Equation 20), we can also determine the momentum radial entropy which has the expression
(64)$$\begin{array}{ccc} R_{q}\left[\gamma_{n,l}^{\left(H\right)}\right] & = & \frac{1}{1-q}\ln\left(\frac{Z^{D}}{\eta^{D}}\frac{K_{n,l}^{2q}}{2^{q(L+2)}}\right)\\ & & +\frac{1}{1-q}\ln\int_{-1}^{1}{\left(1-y\right)}^{lq+\frac{D}{2}-1}{\left(1+y\right)}^{D\left(q-\frac{1}{2}\right)+q\left(l+1\right)-1}C_{n-l-1}^{\left(L+1\right)}{\left(y\right)}^{2q}\;dy \end{array}$$
Then, we use the linearization formula for the (2q)th-power of the Gegenbauer polynomial $C_{n-l-1}^{\left(L+1\right)}\left(y\right)$, which is a particular case of the corresponding formula for Jacobi polynomials [59,60], obtaining the following expression
(65)$$\begin{array}{ccc} R_{q}\left[\gamma_{n,l}^{\left(H\right)}\right] & = & D\ln\frac{Z}{\eta }+\frac{q}{1-q}\ln\left[2\eta \;{\left(\eta -L\right)}_{2L+1}\right]\\ & + & \frac{1}{1-q}\ln{\bar{F}}_{q}\left(D,\eta ,L\right)+\frac{1}{1-q}\ln{\bar{A}}_{q}\left(D,L\right) \end{array}$$
where
(66)$${\bar{A}}_{q}\left(D,L\right)\equiv 2^{2q-1}\;\frac{\Gamma \left(\frac{D}{2}+ql\right)\;\Gamma \left(-\frac{D}{2}+q\left(D+l+1\right)\right)}{\Gamma {\left(\frac{D}{2}+l\right)}^{2q}\;\Gamma \left(q(D+2l+1)\right)}$$
and the symbol ${\bar{F}}_{q}\left(D,\eta ,L\right)$ denotes the following multivariate Srivastava–Daoust function [60,61]
(67)$$\begin{array}{ccc} {\bar{F}}_{q}\left(D,\eta ,L\right) & \equiv & F_{1:1;\ldots ;1}^{1:2;\ldots ;2}\left(\begin{array}{cc} a:b,c;\ldots ;b,c\\ & ;1,\ldots ,1\\ d:e;\ldots ;e \end{array}\right)\\ & & \;=\sum_{i_{1},\ldots ,i_{2q}=0}^{n-l-1}\frac{{\left(a\right)}_{i_{1}+\ldots i_{2q}}}{{\left(d\right)}_{i_{1}+\ldots +i_{2q}}}\frac{{\left(b\right)}_{i_{1}}{\left(c\right)}_{i_{1}}\cdots {\left(b\right)}_{i_{2q}}{\left(c\right)}_{i_{2q}}}{{\left(e\right)}_{i_{1}}\cdots {\left(e\right)}_{i_{2q}}i_{1}!\cdots i_{2q}!} \end{array}$$
with $a=\left(L+\frac{3}{2}\right)q+\frac{D}{2}\left(1-q\right)$, b=−(η−L−1), c=η+L+1, d=q(2L+4), $e=L+\frac{3}{2}$. Note that, when l=n−1 the function $F_{q}\left(D,\eta ,L\right)=1$. In particular, for the ground state n=1, we have that the radial Rényi entropy has the value
$$\begin{array}{ccc} R_{q}\left[\gamma_{1,0}^{\left(H\right)}\right] & = & D\ln\left[\frac{2Z}{D-1}\right]+\frac{q}{1-q}\ln\left[4\;\Gamma (D)\right]\\ & + & \frac{1}{1-q}\ln\left[\frac{\Gamma {\left(\frac{D}{2}\right)}^{1-2q}\;\Gamma \left(D(q-\frac{1}{2})+q\right)}{2\Gamma \left(Dq+q\right)}\right] \end{array}$$
Finally, taking into account the expressions (55) and (57), it only remains to determine the expression of the angular Rényi entropies $R_{q}\left[Y_{l,\{\mu \}}\right]$ defined by Equations (Equation 38)–(Equation 40). The latter quantities can be algebraically calculated as before, obtaining the following analytical expression for the angular Rényi entropy
(68)$$\begin{array}{ccc} \;R_{q}\left[Y_{l,\{\mu \}}\right] & = & \ln\left(2\pi^{\frac{D}{2}}\right)+\frac{1}{1-q}\ln\left[\frac{\Gamma {\left(l+\frac{D}{2}\right)}^{q}}{\Gamma \left(ql+\frac{D}{2}\right)}\frac{\Gamma \left(qm+1\right)}{\Gamma {\left(m+1\right)}^{q}}\right]\\ & & \;+\frac{1}{1-q}\sum_{j=1}^{D-2}\ln\left[B_{q}\left(D,\mu_{j},\mu_{j+1}\right)\;G_{q}\left(D,\mu_{j},\mu_{j+1}\right)\right] \end{array}$$
where
(69)$$\;B_{q}\left(D,\mu_{j},\mu_{j+1}\right)=\frac{1}{{\left[\left(\mu_{j}-\mu_{j+1}\right)!\right]}^{q}}\frac{{\left(2\alpha_{j}+2\mu_{j+1}+1\right)}_{2(\mu_{j}-\mu_{j+1})}^{q}}{{\left(2\alpha_{j}+\mu_{j}+\mu_{j+1}\right)}_{\mu_{j}-\mu_{j+1}}^{q}}\frac{{\left(q\mu_{j+1}+\alpha_{j}+1\right)}_{q(\mu_{j}-\mu_{j+1})}}{{\left(\alpha_{j}+\mu_{j+1}+1\right)}_{\mu_{j}-\mu_{j+1}}^{q}}$$
and
(70)$$\begin{array}{ccc} \;G_{q}\left(D,\mu_{j},\mu_{j+1}\right) & = & F_{1:1;\ldots ;1}^{1:2;\ldots ;2}\left(\begin{array}{cc} a_{j}:b_{j},c_{j};\ldots ;b_{j},c_{j} & \\ & ;1,\ldots ,1\\ d_{j}:e_{j};\ldots ;e_{j} & \end{array}\right)\\ & = & \sum_{i_{1},\ldots ,i_{2q}=0}^{\mu_{j}-\mu_{j+1}}\frac{{\left(a_{j}\right)}_{i_{1}+\ldots i_{2q}}}{{\left(d_{j}\right)}_{i_{1}+\ldots +i_{2q}}}\frac{{\left(b_{j}\right)}_{i_{1}}{\left(c_{j}\right)}_{i_{1}}\cdots {\left(b_{j}\right)}_{i_{2q}}{\left(c_{j}\right)}_{i_{2q}}}{{\left(e_{j}\right)}_{i_{1}}\cdots {\left(e_{j}\right)}_{i_{2q}}i_{1}!\cdots i_{2q}!} \end{array}$$
with $a_{j}=\alpha_{j}+q\mu_{j+1}+\frac{1}{2}$, $b_{j}=-\mu_{j}+\mu_{j+1}$, $c_{j}=2\alpha_{j}+\mu_{j+1}+\mu_{j}$, $d_{j}=2q\mu_{j+1}+2\alpha_{j}+1$ and $e_{j}=\alpha_{j}+\mu_{j+1}+\frac{1}{2}$. Note that the sum becomes finite because $b_{j}$ is a negative integer number, and so ${\left(b_{j}\right)}_{i}=\frac{\Gamma (b_{j}+i)}{\Gamma (b_{j})}=0,\;\forall i>\left|b_{j}\right|$. Let us also highlight that, when $\mu_{j}=\mu_{j+1}$, the function $B_{q}\left(D,\mu_{j},\mu_{j+1}\right)=G_{q}\left(D,\mu_{j},\mu_{j+1}\right)=1$. Moreover, for the particular states with $l=\mu_{1}=\mu_{2}\ldots =\mu_{D-1}=0$, we have the following value
$$\begin{array}{ccc} \;R_{q}\left[Y_{0,\{0\}}\right] & = & \ln\left[\frac{2\;\pi^{\frac{D}{2}}}{\Gamma \left(\frac{D}{2}\right)}\right] \end{array}$$

In conclusion, from Equations (Equation 54), (Equation 61), and (Equation 65), and Equations (Equation 56), (Equation 65), and (Equation 68), we obtain the following expressions for the total Rényi entropies of the *D*-dimensional hydrogenic system in terms of the hyperquantum numbers, the nuclear charge, and the space dimensionality:(71)$$\begin{array}{ccc} R_{q}\left[\rho_{n,l,\{\mu \}}^{\left(H\right)}\right] & = & D\ln\left(\frac{\pi^{\frac{1}{2}}\eta }{2Z}\right)+\frac{q}{1-q}\ln\left(\frac{{\left(\eta -L\right)}_{2L+1}}{2\eta }\right)\\ & & +\frac{1}{1-q}\ln F_{q}\left(D,\eta ,L\right)\;A_{q}\left(D,L\right)+\frac{1}{1-q}\ln\left[\frac{\Gamma {\left(l+\frac{D}{2}\right)}^{q}}{\Gamma \left(ql+\frac{D}{2}\right)}\frac{\Gamma \left(qm+1\right)}{\Gamma {\left(m+1\right)}^{q}}\right]\\ & & +\frac{1}{1-q}\sum_{j=1}^{D-2}\ln\left[B_{q}\left(D,\mu_{j},\mu_{j+1}\right)\;\;G_{q}\left(D,\mu_{j},\mu_{j+1}\right)\right]+\ln2 \end{array}$$
in position space, and
(72)$$\begin{array}{ccc} \;R_{q}\left[\gamma_{n,l,\{\mu \}}^{\left(H\right)}\right] & = & D\ln\left(\frac{\pi^{\frac{1}{2}}Z}{\eta }\right)+\frac{q}{1-q}\ln\left[2\eta \;{\left(\eta -L\right)}_{2L+1}\right]\\ & & \;+\frac{1}{1-q}\ln\left[{\bar{F}}_{q}\left(D,\eta ,L\right){\bar{A}}_{q}\left(D,L\right)\frac{\Gamma {\left(l+\frac{D}{2}\right)}^{q}}{\Gamma \left(ql+\frac{D}{2}\right)}\frac{\Gamma \left(qm+1\right)}{\Gamma {\left(m+1\right)}^{q}}\right]\;\\ & & \;+\frac{1}{1-q}\sum_{j=1}^{D-2}\ln\left[B_{q}\left(D,\mu_{j},\mu_{j+1}\right)\;G_{q}\left(D,\mu_{j},\mu_{j+1}\right)\right]+\ln2 \end{array}$$
in momentum space, respectively. For numerical details and applications to some particular *D*-dimensional hydrogenic states, we refer to [59], where it is also shown that the multidimensional hydrogenic position–momentum Rényi entropy sum, $R_{q}\left[\rho_{n,l,\{\mu \}}^{\left(H\right)}\right]+R_{q^{*}}\left[\gamma_{n,l,\{\mu \}}^{\left(H\right)}\right]$, fulfills the general Rényi-entropy uncertainty relation (Equation 21) [20,21].

## 6. Rényi Entropies of Rydberg Oscillator and Hydrogenic States

The general expressions for the position and momentum Rényi entropies of arbitrary stationary states of multidimensional oscillator and hydrogenic systems are somewhat highbrow and computationally demanding as described in the two previous sections because they require the evaluation of non-trivial multivariate generalized hypergeometric functions. This is especially true for the highly-excited Rydberg states, since then the principal hyperquantum number is very large and the corresponding Rényi integral kernels are highly oscillatory. These extreme states play a very relevant role in the modern quantum technologies, partially due to the fact that then the system is metastable with large lifetimes and only weakly bound. For example, they are atrractive candidates for quantum simulators and quantum sensors of electromagnetic fields in the microwave and teraherz regions.

In this section, we use some powerful methods based on the strong (degree)-asymptotics of Laguerre and Gegenbauer polynomials to determine the Rényi entropies of Rydberg oscillator and hydrogenic systems. These methods are very useful because they do not require to work with any multivariate hypergeometric function, and they allow for obtaining simple and compact expressions for the Rényi entropies, where the physics can be transparently read, at least at first order, in terms of the principal hyperquantum number.

### 6.1. Rényi Entropies of Rydberg Oscillator States

Taking into account the expressions (42) and (Equation 43), the Rényi entropies for the stationary *D*-dimensional oscillator-like state, which are characterized by the hyperspherical quantum numbers $\left(n_{r},l,\left\{\mu \right\}\right)\equiv \left(n_{r},\mu_{1},\mu_{2},\ldots ,\mu_{D-1}\right)$, have the values
(73)$$R_{q}\left[\rho_{n_{r},l,\left\{\mu \right\}}^{\left(O\right)}\right]=\frac{1}{1-q}\log N\left(n_{r},l,D,q\right)+R_{q}\left[Y_{l,\{\mu \}}\right]-\frac{D}{2}\log\omega -\log2$$
where the symbol $N(n_{r},l,D,q)$ denotes the radial Rényi-like functional of Laguerre polynomials given by Equation (Equation 44). To estimate this expression for Rydberg states, we have to determine the asymptotical value $N_{\infty }\left(n_{r},l,D,q\right)$ of the Laguerre functional $N(n_{r},l,D,q)$ in the limit $n_{r}\to \infty$. Then, we have that the Rényi entropies $R_{q}^{\left(Ry\right)}\left[\rho_{n_{r},l,\left\{\mu \right\}}^{\left(O\right)}\right]$ of the Rydberg-like *D*-dimensional harmonic states are given by
(74)$$R_{q}^{\left(Ry\right)}\left[\rho_{n_{r},l,\left\{\mu \right\}}^{\left(O\right)}\right]+\frac{D}{2}\log\omega ≃\frac{1}{1-q}\log N_{\infty }\left(n_{r},l,D,q\right)+R_{q}\left[Y_{l,\{\mu \}}\right],$$
with
(75)$$N_{\infty }\left(n_{r},l,D,q\right)=\lim_{n_{r}\to \infty }\int_{0}^{\infty }{\left({\left[{\tilde{L}}_{n_{r}}^{\left(l+\frac{D}{2}-1\right)}\left(x\right)\right]}^{2}\;w_{l+\frac{D}{2}-1}\left(x\right)\right)}^{q}\;x^{\left(q-1)(1D/2\right)}\;dx$$
This limiting expression can be calculated [72] by means of the theory of the strong asymptotics of Laguerre polynomials [65,110], obtaining when D>2 that
(76)$$N_{\infty }\left(n_{r},l,D,q\right)=\left\{\begin{array}{cc} C\left(\beta ,q\right)\;{\left(2n_{r}\right)}^{(1-q)\;D/2}\;\left(1+o\left(1\right)\right),\; & q\in (0,q^{*})\\ \frac{2}{\pi^{q+1/2}n_{r}^{q/2}}\;\frac{\Gamma (q+1/2)}{\Gamma (q+1)}\;\left(\log n_{r}+O\left(1\right)\right)\;\; & q=q^{*}\\ C_{B}\left(\alpha ,\beta ,q\right)\;n_{r}^{(q-1)D/2-q}\;\left(1+o\left(1\right)\right),\; & q>q^{*} \end{array}\right.\;$$
for $q>0,n_{r}>>1$ and l=0,1,2,…[72]. The symbols $q^{*}:=\frac{D}{D-1}$, $\alpha =l+\frac{D}{2}-1,\;\beta =\left(1-q\right)\left(\alpha -l\right)=\left(q-1\right)\left(1D/2\right)$, and the constants C(β,q) and $C_{B}\left(\alpha ,\beta ,q\right)$ are given by
(77)$$\;C\left(\beta ,q\right):=\frac{2^{\beta +1}}{\pi^{q+1/2}}\;\frac{\Gamma (\beta +1-q/2)\;\Gamma (1-q/2)\;\Gamma (q+1/2)}{\Gamma (\beta +2-q)\;\Gamma (1+q)};\;C_{B}\left(\alpha ,\beta ,q\right):=2\int_{0}^{\infty }t^{2\beta +1}{\left|J_{\alpha }\left(2t\right)\right|}^{2q}\;dt,$$
where $J_{\alpha }\left(z\right)$ denotes the Bessel function of order α[57]). Note that $N_{\infty }\left(n_{r},l,D,q\right)$ does not depend on $n_{r}$, and is equal to $C_{B}\left(\alpha ,\beta ,q\right)$ only when (q−1)D/2−q=0; then, this constancy occurs either when $D=\frac{2q}{q-1}$ or $q=\frac{D}{D-2}$. Moreover, keep in mind in (83) that the explicit expressions of the angular Rényi entropies $R_{q}\left[Y_{l,\{\mu \}}\right]$, which are given by (Equation 38)–(Equation 40), can be analytically found [60,108,109]; however, they are negligible at first order because they do not depend on $n_{r}$ except in the special case $q=\frac{D}{D-2}$. Therefore, for D>2 and $q\neq \frac{D}{D-2}$, the Rényi entropies of the Rydberg-like *D*-dimensional harmonic states are given by
(78)$$R_{q}^{\left(Ry\right)}\left[\rho_{n_{r},l,\left\{\mu \right\}}^{\left(O\right)}\right]≃\frac{1}{1-q}\log N_{\infty }\left(n_{r},l,D,q\right)-\frac{D}{2}\log\omega ,$$
where the symbol $N_{\infty }\left(n_{r},l,D,q\right)$ is given by (Equation 76). Particularly, for the Rydberg states of three-dimensional isotropic harmonic oscillator, the Rényi entropies are discussed monographically in [71]. In addition, for D=2 and D∈[0,2), the Rényi entropies of any *D*-dimensional oscillator-like state of Rydberg type $(n_{r}>>1,l,\left\{\mu \right\}$ are given by Equation (83), where the asymptotical value $N_{\infty }\left(n_{r},l,D,q\right)$ is explicitly given in Refs. [39,72], respectively. Finally, let us also mention that the one-dimensional case is studied in detail in [39] by use of the strong asymptotics of the weighted $L_{q}$-norm of the Hermite polynomials, since these polynomials control the wavefunctions of all the stationary states of the one-dimensional isotropic harmonic oscillator when we use Cartesian coordinates.

Finally, taking into account (Equation 45), one has that the position–momentum Rényi-entropy sum for the Rydberg harmonic states is
(79)$$R_{q}^{\left(Ry\right)}\left[\rho_{n_{r},l,\left\{\mu \right\}}^{\left(O\right)}\right]+R_{p}^{\left(Ry\right)}\left[\gamma_{n_{r},l,\left\{\mu \right\}}^{\left(O\right)}\right]≃\frac{2}{1-q}\log N_{\infty }\left(n_{r},l,D,q\right),$$
which holds for $q>0,n_{r}>>1,\;l=0,1,2,\ldots$ and the asymptotical value $N_{\infty }\left(n_{r},l,D,q\right)$ has been given above. We observe that this sum does not depend on the oscillator strength ω (as expected [111]), and it fulfills not only the general Rényi entropy uncertainty relation for multidimensional quantum systems (Equation 21) [20,21], but also the (conjectured) Rényi entropy uncertainty relation for multidimensional quantum systems subject to a central potential [92].

### 6.2. Rényi Entropies of Rydberg Hydrogenic States

Taking into account the expressions (55) and (Equation 58), the position Rényi entropies for the stationary *D*-dimensional hydrogenic states, which are characterized by the hyperspherical quantum numbers $\left(n,l,\left\{\mu \right\}\right)\equiv \left(n,\mu_{1},\mu_{2},\ldots ,\mu_{D-1}\right)$, have the values
(80)$$R_{q}\left[\rho_{n,l,\{\mu \}}^{\left(H\right)}\right]=\frac{1}{1-q}\;\log\left[\frac{1}{{\left(2\eta \right)}^{q}}N\left(n,l,D,q\right)\right]+R_{q}\left[Y_{l,\{\mu \}}\right]+D\log\frac{\eta }{2Z}$$
where $\eta =n+\frac{D-3}{2}$ and the symbol N(n,l,α,p,β)≡N(n,l,D,q) denotes the Rényi-like functional of Laguerre polynomials given by
(81)$$N\left(n,l,D,q\right)=\int_{0}^{\infty }{\left({\left[{\tilde{L}}_{n-l-1}^{\left(\alpha \right)}\left(x\right)\right]}^{2}\;w_{\alpha }\left(x\right)\right)}^{q}\;x^{\beta }\;dx,$$
with the parameters
(82)α=2L+1=2l+D−2l=0,1,2,…,n−1,q>0andβ=(2D)q+D−1
which guarantee the convergence of integral (Equation 81); i.e., the condition β+qα=2lq+D−1>−1 is always satisfied for physically meaningful values of the parameters. In addition, realize that the angular part $R_{q}\left[Y_{l,\{\mu \}}\right]$ of the Rényi entropy, given by Equations (Equation 38)–(Equation 40), does not depend on the principal quantum number *n* as previously mentioned.

To determine the position Rényi entropies $R_{q}^{\left(Ry\right)}\left[\rho_{n,l,\{\mu \}}^{\left(H\right)}\right]$ for Rydberg hydrogenic states, we have to estimate the asymptotics n→∞ of the Laguerre functional N(n,l,D,q). Indeed, we have that the Rényi entropies of the Rydberg-like *D*-dimensional harmonic states are given by
(83)$$\begin{array}{ccc} R_{q}^{\left(Ry\right)}\left[\rho_{n,l,\{\mu \}}^{\left(H\right)}\right] & ≃ & \frac{1}{1-q}\log N_{\infty }\left(n,l,D,q\right)+R_{q}\left[Y_{l,\{\mu \}}\right]+D\log\frac{\eta }{2Z}-\frac{q}{1-q}\log\left(2\eta \right) \end{array}$$
(84)$$\begin{array}{ccc} & ≃ & \frac{1}{1-q}\log N_{\infty }\left(n,l,D,q\right)+D\log\frac{n}{2Z}-\frac{q}{1-q}\log\left(2n\right) \end{array}$$
with
(85)$$N_{\infty }\left(n,l,D,q\right)=\lim_{n\to \infty }\int_{0}^{\infty }{\left({\left[{\tilde{L}}_{n}^{\left(\alpha \right)}\left(x\right)\right]}^{2}\;w_{\alpha }\left(x\right)\right)}^{q}\;x^{\beta }\;dx$$
with α=2l+D−2andβ=(2D)q+D−1. It remains to evaluate this asymptotical value for all possible values of *D* and q>0. This is a non-trivial task [74] which requires an extensive use of the strong asymptotics of Laguerre polynomials [65,110]. The use of this technique shows that the dominant contribution to the asymptotical value of the integral (Equation 81) comes from different regions of integration defined according to the values (α,β,q), which characterize various asymptotic regimes. Consequently, we must use various asymptotical representations for the Laguerre polynomials at the different scales. See [74] for further details. In particular, for D>2, we have obtained that
(86)$$N_{\infty }\left(n,l,D,q\right)=C\left(\beta ,q\right)\;{\left(2\left(n-l-1\right)\right)}^{1+\beta -q}\;\left(1+\bar{\bar{o}}\left(1\right)\right),$$
for $q\in \left(0,\frac{D-1}{D-2}\right)$ and the constant
(87)$$C\left(\beta ,q\right):=\frac{2^{\beta +1}}{\pi^{q+1/2}}\;\frac{\Gamma (\beta +1-q/2)\;\Gamma (1-q/2)\;\Gamma (q+1/2)}{\Gamma (\beta +2-q)\;\Gamma (1+q)}.$$

In addition, if D=2 (so β=1), we have
(88)$$N_{\infty }\left(n,l,D,q\right)=\left\{\begin{array}{cc} C\left(1,q\right)\;{\left(2\left(n-l-1\right)\right)}^{2-q}\;\left(1+\bar{\bar{o}}\left(1\right)\right)\;, & q\in (0,2)\\ \frac{\ln\left(n-l-1\right)+\underline{\underline{O}}\left(1\right)}{\pi^{2}}\;, & q=2\\ \frac{C_{A}\left(q\right)}{\pi^{q}}\;{\left(4\left(n-l-1\right)\right)}^{\frac{2}{3}\left(2-q\right)}\left(1+\bar{\bar{o}}\left(1\right)\right),\; & q\in (2,5)\\ \left(\frac{C_{A}\left(q\right)}{\pi^{q}4^{2}}+C_{B}\left(\alpha ,1,q\right)\right)\;{\left(n-l-1\right)}^{-2},\; & q=5\\ C_{B}\left(\alpha ,1,q\right)\;{\left(n-l-1\right)}^{-2}\;, & q\in (5,\infty ). \end{array}\right.$$
with α=2l+D−2 and β=(2D)q+D−1 and the constants
(89)$$C_{A}\left(q\right):=\int_{-\infty }^{+\infty }{\left[\frac{2\pi }{\sqrt[{3}]{2}}\;\;{\mathrm{Ai}}^{2}\left(-\frac{t\sqrt[{3}]{2}}{2}\right)\right]}^{q}dt;\;C_{B}\left(\alpha ,\beta ,q\right):=2\int_{0}^{\infty }t^{2\beta +1}{\left|J_{\alpha }\right|}^{2q}\left(2t\right)\;dt.$$
Note that, in these two cases (D>2;D=2), it happens that β>0. For the remaining pairs (D,q) fulfilling that β<0 and β=0, the asymptotics $N_{\infty }\left(n,l,D,q\right)$ has also been found [74]. Moreover, let us highlight that the position Rényi entropies of the three-dimensional hydrogenic system has been monographically studied analytically and numerically [73].

Working similarly in momentum space [75], we have from (Equation 64) that the Rényi entropies are
(90)$$R_{q}\left[\gamma_{n,l,\{\mu \}}^{\left(H\right)}\right]=\frac{1}{1-q}\log I\left(n,l,q,D\right)+R_{q}\left[Y_{l,\{\mu \}}\right]+D\log\frac{Z}{\eta }$$
where the symbol I(n,l,q,D) denotes the following functional of the orthonormal Gegenbauer polynomials:(91)$$\begin{array}{c} I\left(n,l,q,D\right)=\int_{-1}^{1}{\left\{{\left[{\tilde{C}}_{n-l-1}^{\left(l+\frac{D-1}{2}\right)}\left(y\right)\right]}^{2}\;\omega_{l+\frac{D-1}{2}}\left(y\right)\right\}}^{q}{\left(1-y\right)}^{a}{\left(1+y\right)}^{b}\;dy, \end{array}$$
with the weight function $\omega_{\alpha }\left(y\right)={\left(1-y^{2}\right)}^{\alpha -\frac{1}{2}}$ and the parameters a≡a(q,D), b≡b(q,D) given by
(92)$$a:=\left(1-q\right)\left(\frac{D}{2}-1\right),\;b:=-\left(1-q\right)\left(\frac{D}{2}+1\right)+q.$$
Now, for the Rydberg states, the last three expressions show that the Rényi entropies of the *D*-dimensional hydrogenic system can be expressed as
(93)$$R_{q}^{\left(Ry\right)}\left[\gamma_{n,l,\{\mu \}}^{\left(H\right)}\right]≃\frac{1}{1-q}\log I_{\infty }\left(n,l,q,D\right)+R_{q}\left[Y_{l,\{\mu \}}\right]+D\log\frac{Z}{\eta },$$
where the symbol $I_{\infty }\left(n,l,q,D\right)$ denotes the limiting expression
(94)$$\begin{array}{ccc} I_{\infty }\left(n,l,q,D\right) & = & \lim_{n\to \infty }I\left(n,l,q,D\right)\\ & = & \lim_{n\to \infty }\int_{-1}^{1}{\left\{{\left[{\tilde{C}}_{n-l-1}^{\left(l+\frac{D-1}{2}\right)}\left(y\right)\right]}^{2}\;\omega_{l+\frac{D-1}{2}}\left(y\right)\right\}}^{q}{\left(1-y\right)}^{a}{\left(1+y\right)}^{b}\;dy, \end{array}$$
which has ben recently estimated [75] at first order, obtaining
(95)$$\begin{array}{ccc} I_{\infty } & ≍\;\;n^{q-2(b+1)};\;\mathrm{for} & \frac{1}{2}\;\;\frac{D}{l+D+1}<\;q\;<\frac{D}{D+3}, \end{array}$$
(96)$$\begin{array}{ccc} I_{\infty }\;\; & ∼\;\;\log n;\;\;\mathrm{for}\; & q=\frac{D}{D+3}, \end{array}$$
(97)$$\begin{array}{ccc} I_{\infty }\;\; & =\;\;c(q,D);\;\;\mathrm{for}\; & \frac{D}{D+3}<q<\frac{D}{D-1}, \end{array}$$
(98)$$\begin{array}{ccc} I_{\infty }\;\; & ∼\;\;\log n;\;\;\mathrm{for}\; & \;\;\;q=\frac{D}{D-1}, \end{array}$$
(99)I∞≍nq−2(a+1)forDD−1<q<DD+3,
where the symbol c(q,D) denotes the constant
$$c\left(q,D\right)=\frac{2^{a+b+1}}{\pi^{q+1}}\frac{\Gamma \left(q+\frac{1}{2}\right)\Gamma \left(\frac{1}{2}\right)}{\Gamma (q+1)}\frac{\Gamma \left(a-\frac{q}{2}+1\right)\Gamma \left(b-\frac{q}{2}+1\right)}{\Gamma (a+b-q+2)}.$$
Taking into account this result and that the angular part $R_{q}\left[Y_{l,\{\mu \}}\right]$ given by Equation (Equation 59) does not depend on *n*, the momentum Rényi entropies of the Rydberg hydrogenic states have the following asymptotical behavior
(100)$$R_{q}^{\left(Ry\right)}\left[\gamma_{n,l,\{\mu \}}^{\left(H\right)}\right]≃\frac{1}{1-q}\log I_{\infty }\left(n,l,q,D\right)+D\log\frac{Z}{n},$$
and finally
(101)$$R_{q}^{\left(Ry\right)}\left[\gamma_{n,l,\{\mu \}}^{\left(H\right)}\right]\left\{\begin{array}{cc} ∼-\frac{3q}{1-q}\log\;n\;\; & q\in (q_{*},q^{*})\\ =-D\log\;n+\frac{1}{1-q}\log\log\;n\;+\;O\left(1\right),\; & q=q^{*}\\ =-D\log\;n+\frac{1}{1-q}\log c\left(q,D\right)+o\left(1\right)\;\; & q\in (q^{*},q^{+})\\ =-D\log\;n+\frac{1}{1-q}\log\log\;n\;+\;O\left(1\right),\; & q=q^{+}\\ ≍(-2D-\frac{q}{1-q})\log\;n\;\; & q>q^{+} \end{array}\right.\;$$
for n>>1,l=0,1,2,… and D>0. The symbols $q_{*}:=\frac{1}{2}\;\;\frac{D}{l+D+1}$, $q^{*}:=\frac{D}{D+3}$ and $q^{+}:=\frac{D}{D-1}$. Moreover, $q_{*}=q^{*}$ for D=1 and l=0, and $q_{*}<q^{*}$ for D>1. Note that the momentum Rényi entropies of Rydberg hydrogenic states grow logarithmically with *n* for all $q>q_{*}$. Finally, it is interesting to remark from Equations (84)–(Equation 88) and (Equation 101) that the total position–momentum Rényi entropies $R_{q}^{\left(Ry\right)}\left[\rho_{n,l,\{\mu \}}^{\left(H\right)}\right]+R_{p}^{\left(Ry\right)}\left[\gamma_{n,l,\{\mu \}}^{\left(H\right)}\right]$ of the Rydberg multidimensional hydrogenic states do not depend on the nuclear charge *Z* of the system (as expected [111]) and fulfills not only the general Rényi entropy uncertainty relation for multidimensional quantum systems (Equation 21) [20,21], but also the (conjectured) Rényi entropy uncertainty relation for multidimensional quantum systems subject to a central potential [92].

## 7. Conclusions

First, we have shown the rigorous bounds for the Rényi entropies of general and central-potential multidimensional quantum systems, beginning with the associated entropic uncertainty relations. Then, we have analytically shown and discussed the exact determination of the Rényi entropies of the multidimensional oscillator and hydrogenic states in terms of the potential strength, the spatial dimensionality, and the state’s hyperquantum numbers. We have used some linearization techniques of powers of hypergeometric orthogonal polynomials to solve the involved Rényi-like integral functionals of the Laguerre and Gegenbauer polynomials which control the wavefunctions of the quantum states.

We have obtained the analytical expressions for the Rényi entropies of the stationary states of both oscillator and hydrogenic systems by means of some multivariate hypergeometric functions. These expressions are somewhat highbrow and computationally demanding, especially when the principal hyperquantum number becomes large because then the Rényi integral kernel is highly oscillatory; this occurs, e.g., for the relevant class of highly-excited Rydberg multidimensional states: they are promising elements to store and manipulate quantum information for both quantum computation and simulation among many other applications due to their extraordinary properties (see, e.g., [80,81,82,83,84]).

Finally, the Rényi entropies for the highly-excited Rydberg states of the multidimensional oscillator and hydrogenic states are analytically calculated by using some powerful techniques of approximation theory which are based on the strong (degree) asymptotics of the Laguerre and Gegenbauer polynomials. This method allows for the exact determination of the dominant contribution to the Rényi entropies of both multidimensional oscillator and hydrogenic states in a compact form, which shows the dependence of these entropic quantities on the spatial dimensionality and the principal hyperquantum number in a simple and transparent way.

## Data Availability

Not applicable.

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
