# Peer review of "Rényi Entropies of Multidimensional Oscillator and Hydrogenic Systems with Applications to Highly Excited Rydberg States"

_entropy, 2022, doi:10.3390/e24111590_

Round 1

Reviewer 1 Report

Despite the fact that I did not review all the calculations in detail and did not detect inconsistencies, I consider that this article deserves its publication.

I have only one comment: I suggest to add in the instruction a brief description about the meaning of the parameter q^*, similar to the comment about q. Perhaps clarify the role of q and q^* in each of the two complementary spaces.

Author Response

First of all, I thank this referee for his positive considerations
about my work that lead him to conclude that my article deserves its
publication.

On your final comment on the Rényi parameter: it goes without saying
that the role of q* in momentum space is the same as that of q in
position space.

Reviewer 2 Report

In this paper the author has obtained analytically Renyi entropies of D dimensional general and central potential systems. As examples, harmonic oscillator and Coulombic system have been considered. Subsequently rigorous bounds on Renyi entropies have been obtained. I believe the results are correct and the paper has been written in a lucid fashion. I recommend publication of the paper in the present form.

Author Response

Dear Reviewer,

Thank you very much for your positive consideration about the contents of my paper, and for your recommendation of its publication.

Kind regards.

Reviewer 3 Report

This paper presents lower and upper bounds for the Renyi entropies of general and central-potential quantum systems, as well as the associated entropic uncertainty relations. 

The topic of Renyi entropy (and related quantities) is an interesting one, as it characterizes the uncertainty measures in a much better way than the Heisenberg uncertainty relation. This is due to the fact that it takes into account all the moments of the distribution so it does not give a large weight to the tails of the distribution. As the paper aptly points out,  these entropies have allowed to gain a much deeper knowledge of many scientific and technological phenomena.

The author has contributed in a very active way. The results presented here are sound and are presented in a very professional way. I have no technical comments to address, because the paper, to the best of my knowledge, is correct.

However, after reading such a long manuscript, clogged with complicated formulas, I have the feeling that all this stuff is just a mathematical exercise. My final question is: What for? D-dimensional central potentials and exotic systems like these are truly necessary to understand the behavior of Renyi entropy? I think the author should give some motivation about the physical interest of these results in otder to make his results more appealing.

Author Response

Dear reviewer,

Thank you very much for your positive considerations about my paper and its results.

About your "mathematical-exercise feeling":  The mathematical formulas discussed in the paper are not a simple exercise but, on the contrary, they require a deep knowledge of

 1.- non-trivial multivariate hypergeometric functions and a formidable use               of non-trivial linearization techniques of powers of orthogonal                 polynomials to solve the Rényi integral functionals of the quantum systems,

2.- some powerful asymptotical methods of approximation theory which are able to calculate in a compact and transparent form the dominant term of the asymptotics of the Rényi-like functionals of Laguerre and Gegenbauer polynomials involved in the determination of the quantum Rényi entropies.  They are not yet so well known in the mathematical and physical literature, because they have been developed in the last few years. 

Kind regards.